# Effects of Patient Education on Pain and Function and Its Impact on Conservative Treatment in Elderly Patients with Pain Related to Hip and Knee Osteoarthritis: A Systematic Review

**DOI:** 10.3390/ijerph19106194

**Published:** 2022-05-19

**Authors:** Pierluigi Sinatti, Eleuterio A. Sánchez Romero, Oliver Martínez-Pozas, Jorge H. Villafañe

**Affiliations:** 1Department of Physiotherapy, Faculty of Sport Sciences, Universidad Europea de Madrid, 28670 Villaviciosa de Odón, Spain; plgsinatti@gmail.com; 2Musculoskeletal Pain and Motor Control Research Group, Faculty of Sport Sciences, Universidad Europea de Madrid, 28670 Villaviciosa de Odón, Spain; oliver.martp@gmail.com; 3Department of Physiotherapy, Faculty of Health Sciences, Universidad Europea de Canarias, 38300 La Orotava, Spain; 4Musculoskeletal Pain and Motor Control Research Group, Faculty of Health Sciences, Universidad Europea de Canarias, 38300 Tenerife, Spain; 5Escuela Internacional de Doctorado, Department of Physical Therapy, Occupational Therapy, Rehabilitation and Physical Medicine, Universidad Rey Juan Carlos, 28933 Alcorcón, Spain; 6IRCCS Fondazione Don Carlo Gnocchi, Piazzale Morandi 6, 20148 Milan, Italy

**Keywords:** education, conservative treatment, pain, function, osteoarthritis, elderly

## Abstract

(1) Background: Patient education (PE), exercise therapy, and weight management are recommended as first-line interventions for hip and knee osteoarthritis (OA). Evidence supporting the effectiveness of exercise therapy and weight management in people with lower-limb OA has been synthesized in recent studies. However, according to the Osteoarthritis Research Society International, PE is often considered a standard of care and the inclusion of this as a first-line intervention for people with knee OA in clinical practice guidelines is often supported by limited evidence. The aim of this review is to evaluate the effects of PE on pain and function and how it impacts on conservative treatment. (2) Methods: This is a literature review of studies investigating the effect of patient education on pain and function and its impact on conservative treatment in elderly patients with pain related to hip and knee OA. PRISMA guidelines were followed during the design, search, and reporting stages of this review. The search was carried out in the PubMed database. (3) Results: A total of 1732 studies were detected and analyzed by performing the proposed searches in the detailed database. After removing duplicates and analyzing the titles and abstracts of the remaining articles, 20 studies were ultimately selected for this review. Nineteen of these twenty articles showed positive results in pain or function in patients with pain related to hip and knee OA. (4) Conclusions: PE seems to be effective in reducing pain and improving function in patients with pain related to hip and knee OA. Furthermore patient education seems to positively impact the conservative treatment with which it can be associated.

## 1. Introduction

Osteoarthritis (OA) is a degenerative alteration of the articular cartilage frequently characterized by pain, deformity, instability, and functional limitation. It generally affects elderly patients and is a leading cause of disability in the adult population worldwide [1,2]. According to the World Health Organization (WHO) over 343 million people are affected by some form of OA, and the incidence is higher in women than in men [1,2,3,4]. Furthermore, the prevalence of the disease is greater in Europe and the USA than in other parts of the world [1,3,4]. The healthcare burden related to OA is growing and in many developed countries is considered unsustainable. For example in Spain and Italy—two of the countries with the longest life expectancy in Europe—the average annual cost for OA medications per patient was estimated between €1000 and 1500 per year [5,6]. Globally costs take on an even more impressive aspect. In the UK, the expenses for non-steroidal anti-inflammatory drugs were estimated at around £20 million; the cost of arthroscopic surgery for OA was estimated to be £1.34 million; hip and knee replacements were estimated to exceed £850 million; and indirect costs from OA, such as social and community services, caused a significant loss of economic production of over £3.2 billion. In addition, in France, OA is a conspicuous public burden, with direct costs of about 1.7% of the expenses of the French Health system, staying just below €2 billion [5]. The American College of Rheumatology and the Arthritis Foundation recommend weight management, exercise therapy, some types of bracing (tibiofemoral), and patient education (PE) as first-line interventions for lower-limb OA [7]. Evidence supporting the effectiveness and cost-effectiveness of exercise therapy, weight management, and nutritional therapy in people with lower-limb OA has been synthesized in recent studies [7,8,9,10,11,12,13]. However, according to The Osteoarthritis Research Society International, PE is often considered a standard of care [8], and the inclusion of this as a first-line intervention for people with knee OA in clinical practice guidelines is often supported by limited and no specific evidence. In fact, there is a well-established body of literature supporting the use of education in some conditions such as OA elsewhere in the body and chronic pain [7,8,14], but there are no recent high-quality studies specifically evaluating the effectiveness of patient education on pain and function outcomes in people with hip and knee OA, and how it affects the conservative treatment to which it could be associated. A recent review published in the Journal of Physiotherapy in 2021 by Goff et al., analyzed the effect of PE as a standalone intervention or combined with other interventions for people with OA [15]. However, this review did not include participants with hip OA. It only included patients with knee OA. Furthermore, the authors analyzed the effects of education on joint-related pain, function, and psychological variables of the patient, but did not evaluate the impact that this intervention may have on the conservative treatment with which it is associated. The results of this review denoted that PE may reduce pain and improve function compared with usual care in people with knee OA. Additionally, Goff et al., showed that PE with physical exercise should be encouraged considering clinically important improvements in function compared with patient education alone. In 2012, Kron et al., published a high-quality evidence review evaluating the effects of PE on pain and function in patients with OA [16]. This study did not distinguish knee OA and other arthritic conditions, and reported little to no benefit of PE compared with providing information only, usual care only, or no treatment. A more recent review realized by Gay et al., in 2016 highlighted the role of PE in exercise and weight loss programs in the treatment of hip and knee OA [10]. This review only examined the impact of education on physical activity and weight loss, not on other conservative treatments, and did not analyze the effect of PE on pain and function. According to recent guidelines, education can positively impact on conservative treatment, enhancing compliance to exercise and weight loss programs, thereby improving their long-term benefits. There is a lack of up-to-date evidence synthesis for PE impact on conservative treatment and its effects on pain and function in people with hip and knee OA to inform guidelines and practice. PE may have relevance for the management of hip and knee OA, and it may positively influence conservative treatment. Across previous studies, it has been shown that there are improvements with consistent benefits and minimal risks [7,8,14]. Education can involve some form of communication and consists of improving patient knowledge, and developing self-management and life skills, which are conducive to individual—and community—health. It can be performed in many different ways with sessions on self-efficacy and self-management, skill-building (goal-setting, problem-solving, positive thinking), education about the disease and about medication effects and side effects, joint protection measures, exercise goals and approaches [17]. The aim of this review is to estimate the effects of PE on pain and function and how it impacts conservative treatment in elderly patients with pain related to hip and knee OA. This article could provide evidence for the inclusion of PE as a first-line intervention for people with knee and hip OA in practice guidelines, benefiting patients, researchers, and health professionals.

## 2. Materials and Methods

This is a systematic literature review of studies investigating the effectiveness of education on conservative treatment in patients with pain related to hip and knee OA. The design, search, and reporting of this systematic review followed the Preferred Reporting Items for Systematic Reviews and Meta-Analyses (PRISMA) statement [18]. The protocol of this systematic review was registered in PROSPERO (ID: CRD42022300133) before starting the article. There are no discrepancies between the registered protocol and this manuscript.

### 2.1. Search Strategy

Our literature search aimed at identifying all available studies that evaluated the effects of education, combined or not with other conservative treatments, on pain and function in people with pain-related OA. The systematic search of the articles was carried out by a single reviewer (PS). The PubMed database was used as search engines and the search string was: (((((education) OR (conservative treatment)) AND ((osteoarthritis) OR (osteoarthrosis))) AND ((lower limb) OR (knee) OR (hip))) AND ((elderly) OR (older patients) OR (older adults))) AND ((pain) OR (function)).

### 2.2. Eligibility Criteria

The studies included in this systematic review met the following criteria: (a) no date restrictions and free or paid availability; (b) experimental and observational articles; (c) elderly men and women (age > 50) with either clinical or radiographically diagnosed hip or knee OA related pain; (d) any form of educational intervention, combined or not with other conservative treatment and compared with any conservative non-pharmacological intervention; and (e) assess pain and/or functionality. Studies in which education was provided to the control group were excluded as they did not adequately analyze the effects of patient education but the effect of the intervention in the experimental group. In addition, all repeated articles, case reports, letters to editor, pilot studies, editorials, technical notes, and review articles were excluded from the analysis.

### 2.3. Data Extraction

All relevant articles from the database were identified by one reviewer (PS) who conducted the search and the data extraction by reading titles and the abstracts of each article resulting from the search string. Subsequently, once this first step of the selection process was completed, the exclusion of duplicate studies and those that after reading the full text were not related to the study question was agreed upon by two researchers (PS and OMP) in a parallel and consensual manner. A standardized form was used by PS and OMP to extract and collect information on the characteristics of the studies (study design, authors, year of publication), characteristics of participants (study population, number of subjects), assessment and follow-up timing, characteristics of the interventions, clinical outcome measures, and reported findings. Finally, if necessary, in the event that there was disagreement in the inclusion or exclusion of articles, a protocol was designed so that another two researchers (JHV and EASR) would act as decision-makers.

### 2.4. Quality Assessment

The evaluation of the methodological quality was carried out using the PEDro Scale [19]. This analysis instrument has been reported to be a valid and reliable instrument to measure the methodological quality of clinical intervention trials. It is made up of 11 items, each one valued with one point that allows evaluating whether randomized clinical trials may have sufficient internal validity (criteria 2–9) and sufficient statistical information to make their results interpretable (criteria 10–11). These parameters were assessed by PS and OMP and all disagreements were resolved until consensus was reached.

### 2.5. Risk of Bias Assessment

The risk of bias analysis of the randomized clinical trials was carried out by PS and OMP using the Cochrane risk-of-bias tool for randomized trials (RoB 2.0) [20]. This tool assesses the methodology used by researchers in the development of a clinical trial, scoring individually the presence of the following biases. From the score obtained, it was interpreted taking into consideration that a risk of bias “low” implies that the bias committed is unlikely to significantly alter the results, the risk of bias “some concerns”, that there are some doubts about the results, while the “high” risk of bias would be indicative of weak confidence in the results obtained. These parameters were assessed by PS and OMP and all disagreements were resolved until consensus was reached.

The risk of bias analysis of the randomized clinical trials was carried out by PS and OMP using the Cochrane risk-of-bias tool for randomized trials (RoB 2.0) [20]. This tool assesses the methodology used by researchers in the development of a clinical trial, scoring individually the presence of the following biases. From the score obtained, it was interpreted taking into consideration that a risk of bias “low” implies that the bias committed is unlikely to significantly alter the results, the risk of bias “some concerns”, that there are some doubts about the results, while the “high” risk of bias would be indicative of weak confidence in the results obtained. These parameters were assessed by PS and OMP and all disagreements were resolved until consensus was reached.

## 3. Results

### 3.1. Study Selection

Originally 1731 studies were identified through the database search. Once duplicates were removed and the titles and abstracts of all remaining unique articles were analyzed, 90 full-text articles were analyzed to verify their eligibility for inclusion in the present study. Seventy of these articles were excluded. Twenty studies were finally selected for this review. The flow of studies through the review process can be found in Figure 1.

### 3.2. Quality Assessment

The PEDro Scale was used to assess the quality of all clinical trials, randomized and non-randomized. Two of the articles reviewed were of high quality (score 9–10) [21,22], Fifteen were of good quality (score 6 and 8) [23,24,25,26,27,28,29,30,31,32,33,34,35,36,37] and three were of fair quality (score 4–5) [38,39,40]. The results of PEDro scale can be found in Table 1.

### 3.3. Risk of Bias within and across the Studies

The risk of bias analysis of the twenty randomized clinical trials included in the review was carried out using the Cochrane risk-of-bias tool for randomized trials (RoB 2.0). Most of the articles included in the review (9) were at “low risk” of bias [21,22,23,26,28,30,32,36,37]. In the remaining twelve, six were in “some concerns” [24,27,29,37,38,39], and six “high risk” [25,31,33,34,35,40]. Results of risk of bias are summarized in Table 2.

### 3.4. Data from Studies

#### 3.4.1. Effects of Patient Education on Pain

In the twenty articles included in the review, eighteen analyzed the effect of patient education on pain in subjects with knee and/or hip OA. Most of these, showed positive results for PE on pain. In fact, there are sixteen studies in which PE, used as a standalone intervention or in combination with other conservative treatments, produced significant differences in pain scores compared to the control group. In nine of these articles, education was used alone in the experimental group, showing positive results [23,24,25,26,29,34,35,36,38]. The remaining six studies used PE in combination with other conservative treatments: in five articles, education was applied in combination with exercise [21,28,30,33,40], and only in one study was it used with manual therapy [31]. There are only two articles in which education does not appear to produce positive effects on pain: In Allen et al., (2019) PE has been used as a standalone, in the form of a PCST program compared to a waitlist control group of 248 people with hip or knee OA [32]. Findings showed that the PCST program did not significantly reduce pain. In fact, there were no significant between-group differences in WOMAC pain score at 3 (20.63 [95% CI 21.45, 0.18]; *p* = 0.128) or 9 months (20.84 [95% CI 21.73, 0.06]; *p* = 0.068). The remaining study used education in association with another intervention. In Bennell et al., (2014), PE was used in combination with manual therapy, home exercise, and gait aids, compared to a sham intervention [22]. The study included 102 patients with hip OA and pain: 49 patients in the active group and 53 in the sham group experienced 12 weeks of intervention and 24 weeks of follow-up. The between-group differences for improvements in pain were not significant. For the active group, the baseline mean visual analog scale score was 58.8 mm (13.3) and the after-intervention score was 40.1 mm (24.6); for the sham group, the initial score was 58.0 mm (11.6), and the after-intervention score was 35.2 mm (21.4). The mean difference was 6.9 mm favoring the sham treatment (95% CI, −3.9 to 17.7).

#### 3.4.2. Effects of Patient Education on Function

The results of the studies included in this review were predominantly positive for the use of education in the function variable. In the twenty articles included, seventeen analyzed the effect of patient education on function in subjects with knee and/or hip OA. Of these, only three articles did not show positive results for education on function [22,23,28]. One of them was the article realized by Allen et al., in 2010 that examined the effectiveness of a telephone-based osteoarthritis self-management intervention in 515 patients with hip or knee osteoarthritis in a primary care setting compared with health education (attention control), and usual care control groups [23]. The OA self-management intervention involved educational materials and 12 monthly telephone calls to support individualized goals. The health education intervention consisted of educational materials not inherent to OA and 12 monthly telephone calls related to general health topics. In conclusion, the article realized by Allen et al., showed that a telephone-based OA self-management program did not produce statistically significant improvements in function in patients with OA, compared with a control group. Another article that did not show the positive effects of education on function was the one realized by Lawford et al., in 2018 [28]. This article explored the effect of an internet-delivered intervention on function and pain in people with knee OA. The authors realized a RCT comparing internet-delivered exercise, education, and pain coping skills training to internet-delivered education alone in 148 patients with knee OA. No differences between groups in terms of function were detected at any follow-up. The last of the few articles that showed negative results for function was the one realized by Bennel et al., in 2014 [22]. The authors realized a study with the aim to determine the efficacy of physical therapy on pain and physical function in 102 people with pain and hip OA. Forty-nine patients in the active group and fifty-three in the sham group underwent 12 weeks of intervention and 24 weeks of follow-up. Patients attended 10 treatment sessions over 12weeks. Active treatment included PE, manual therapy, and exercise, while the sham treatment consisted of inactive ultrasound and inert gel. The function scores were not significantly different between groups. The baseline mean (SD) physical function score for the active group was 32.3 (9.2) and the week-13 score was 27.5 (12.9) units, whereas the baseline score for the sham treatment group was 32.4 (8.4) units and the week-13 score was 26.4 (11.3) units, for a mean difference of 1.4 units in favor of sham (95% CI, −3.8 to 6.5) at week 13. Among adults with painful hip OA, the active intervention described above did not result in greater improvement in function compared with sham treatment. All the remaining fourteen articles included in the review that evaluated the effectiveness of education in function in patients with osteoarthritis showed positive results. Most of these, seven, performed an intervention that combined education with a form of physical exercise [21,27,30,33,37,39,40]. Six articles carried out an analysis with patient education as a standalone intervention [25,26,32,34,35,36] and one study combined education with manual therapy [31]. The basic characteristics of the studies included in the review are summarized in Table 3.

## 4. Discussion

This review provides a comprehensive synthesis of evidence related to patient education for hip and knee OA, which can inform guidelines, clinical practice, and future research. To date, 20 studies have been carried out to analyze the effect of patient education on pain and function and how it impacts conservative treatment in elderly patients with pain related to hip and knee OA. Among the articles reviewed, patient education was applied in four different ways: in combination with exercise [21,27,28,30,33,37,39,40], in combination with manual therapy [31], in combination with both [22], and as a standalone intervention [23,24,25,26,29,32,34,35,36,38]. The results obtained are discussed below.

### 4.1. Discussion on the Effects of Patient Education on Pain

Eighty-four percent of the studies that analyzed the effect of education on pain showed significantly positive effects compared to the control group. Among all the articles that analyzed pain outcomes, except the one realized by Allen et al., in 2019 [32], all the studies in which education was applied as a standalone intervention showed positive effects on pain compared to the control group [23,24,25,26,29,34,35,36,38], with a percentage of 90%. The same applies to studies where education was applied in combination with exercise: five of six studies showed significant positive effects on pain [21,28,30,33,40]. Only in Lluch et al., [31], education was associated solely with manual therapy. In this study, two types of education associated with a manual therapy technique for the lower limb were compared. The two types of intervention showed no differences between them in pain, but both improve this outcome. As evidenced by the studies analyzed in this review where education was applied to the pain outcome, it is possible to state that it can be effective alone and in combination with exercise or manual therapy in improving pain in subjects with pain related to hip or knee OA.

### 4.2. Discussion on the Effects of Patient Education on Function

About 82% of the studies that analyzed the effect of education on function, showed significant positive effects. With the exception of the study realized by Lawford et al., [28], all studies in which education was associated with exercise showed significant effects on function [21,27,30,33,37,39,40], with a percentage of 87.5%. All but one [23] study where education was used as a standalone intervention showed significant positive effects on function [25,26,32,34,35,36], for a total of 85.7%. The only study in which education is associated solely with manual therapy demonstrated positive effects [31]. As evidenced by these results, it is possible to state that education can be recommended alone and in combination with exercise or manual therapy to improve function in subjects with pain related to hip or knee OA.

### 4.3. Discussion on the Impact of Patient Education on Conservative Treatment

One of the aims for which this review was carried out was to evaluate the impact of therapeutic education on the conservative treatment with which it is associated. The studies where was it possible to evaluate the impact of PE on the conservative treatment were those in which the experimental group consisted of conservative treatment and PE and was compared with a control group composed solely of subjects to whom only the conservative treatment present in the experimental group was applied. Four studies present in the review used this approach and the results are discussed below [21,26,27,37].

In the study realized by Somers et al., 2012, the authors examined the efficacy of a combined PCST and lifestyle behavioral weight management (BWM) intervention in overweight patients with OA [26]. Two hundred and thirty-two patients were randomly assigned to one of the following groups: (1) PCST + BWM; (2) PCST alone; (3) BWM-only; or (4) standard care control. Assessments of pain, physical disability, psychological disability, and body weight were collected at four time points: baseline, post-treatment (6 months), and after the completion of treatment at 6 and 12 months. Patients randomized to group 1 composed of PCST + BWM demonstrated significantly better outcomes in terms of pain, physical disability, stiffness, activity, weight self-efficacy, and weight when compared to the other three groups (*p* < 0.05). The group composed of PCST + BWM also did significantly better than at least one of the other conditions (i.e., PCST-only, BWM-only, or standard care) in terms of arthritis self-efficacy, pain catastrophizing, and psychological disability. In conclusion, interventions teaching overweight and obese OA patients with PCST and weight management simultaneously may provide more comprehensive long-term benefits. The results of the study, therefore, highlight a significantly positive impact of education on the conservative treatment with which it is associated as the PCST + BWM group resulted in better outcomes than the group composed of BWM only [26].

Arnold et al., in 2010, evaluated the effect of aquatic exercise and education on fall risk factors in older adults with hip osteoarthritis (OA) [37]. Seventy-nine adults, 65 years of age or older with hip OA and at least one fall risk factor, were randomly assigned to one of three groups: aquatics and education (AE; aquatic exercise twice a week with once-a-week group education), aquatics only (A; 2-week aquatic exercise) and control (C; usual activity). Balance, falls efficacy, dual-task function, functional performance (chair stands), and walking performance were measured pre- and post-intervention or control period. In the results, there was a significant improvement in fall risk factors (full-factorial MANCOVA, baseline values as covariates; *p* = 0.038); AE improved in falls efficacy compared with C and in functional performance compared with A and C. In conclusion, Arnold et al., showed that the combination of aquatic exercise and education was effective in improving fall risk factors in older adults with arthritis compared to aquatic exercise only, demonstrating a significantly positive impact of education on the conservative treatment [37].

Arnold et al., in 2011, explored differences in fall-risk outcomes in 54 older patients with hip OA with higher versus lower levels of falls efficacy and evaluated the relationship between initial falls-efficacy status and changes in fall risk factors following two different interventions [27]. Patients received two different types of intervention for 11 weeks: aquatic exercise twice a week plus education once a week or aquatic exercise only, twice a week. Patients in the exercise plus education group with low baseline falls efficacy demonstrated significantly (*p* < 0.05) greater improvement in balance and falls efficacy compared to patients in the exercise-only group with high baseline falls efficacy. In the exercise plus education group only, baseline falls-efficacy status was significantly (*p* < 0.05) correlated with positive balance and falls-efficacy change scores (Spearman rank r 1⁄4 0.45 and 0.63, respectively). The authors concluded that individuals with one or more fall-risk factors and low falls efficacy may benefit more from receiving a treatment that combines exercise with education than receiving an intervention with exercise only [27]. Bennell et al., in 2018, analyzed the effects of an intervention composed of an internet-based PCST program plus home exercise for people with hip OA [21]. One hundred and forty-four people were randomized to either the PCST group or control group. In the first 8 weeks, the PCST group received online education and PCST while the control group received online education only. From weeks 8–24, both groups realized home exercise. Assessments were performed at baseline, 8, 24, and 52 weeks. Primary outcomes were hip pain on walking and physical function. There were no significant between-group differences in primary outcomes at week 24, with both groups showing clinically-relevant improvements. At week 8, the PCST group had greater improvements in function, pain coping, and global improvement than the control group. Greater pain-coping improvements persisted at 24 and 52 weeks. This article showed that the experimental group which had a greater and more complete education intervention due to the combination of PCST and education improved more at 8, 24, and 52 weeks than the comparison composed of education and exercise only [21].

In conclusion, the four articles listed above showed how education positively affects the conservative treatment to which it is applied [21,26,27,37]. In fact, each of these studies showed that an intervention consisting of education associated with conservative treatment produces better results than an intervention composed of a single conservative treatment. In accordance with these results, there are other studies in this review: in fact, excluding the studies described above, there are seven out of sixteen studies in which an intervention composed of education with conservative treatment was applied. In each of these studies, except one [22], there was an improvement in at least one of the pain or function outcomes, showing remarkable effectiveness of this association [27,28,31,33,39,40].

There is a well-established body of literature supporting the use of education in OA elsewhere in the body [1,7,8,14] and the results of this review may provide evidence in favor of the effectiveness of PE in knee and hip OA. Findings could support the inclusion of education as a first-line intervention for people with knee and hip OA in the practice guidelines, benefiting patients and health professionals. In fact, education, a low-cost and minimal risk intervention, could be included in daily clinical practice as a standalone intervention or associated with conservative treatment in those patients with OA to improve pain and function. Future research could investigate what kind of PE could be good for patients mostly and how it would be the most effective to apply.

### 4.4. Limitations

Some possible limitations of the present review could have been reduced by utilizing using multiple databases to search for articles. In addition, it should be noted that the methodological quality of the studies selected for this review was not always of high quality.

## 5. Conclusions

The review findings indicated that PE used as a standalone intervention may reduce pain and improve function compared with usual care. PE could improve pain and function even when applied combined with others conservative treatments. Furthermore, results suggested that PE could positively impact the treatment it is associated with. Findings suggested that combining a conservative treatment, such as exercise therapy or manual therapy, with PE should be encouraged considering statistically superior and clinically important improvements in patient outcomes compared with the conservative treatment alone.

## Figures and Tables

**Figure 1 ijerph-19-06194-f001:**
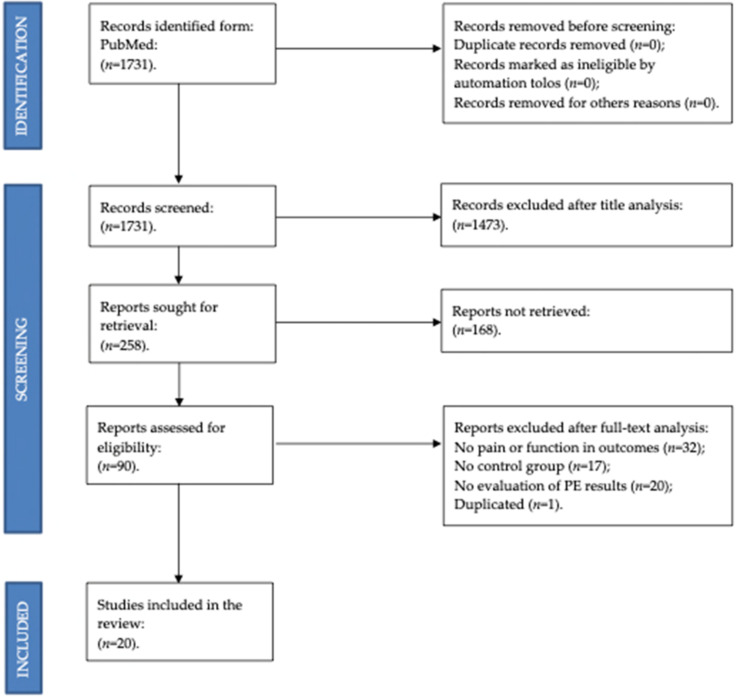
PRISMA flow diagram.

**Table 1 ijerph-19-06194-t001:** Pedro scale.

	1	2	3	4	5	6	7	8	9	10	11	Score
Keefe et al., (1996)	Yes	Yes	No	Yes	No	No	No	Yes	No	Yes	Yes	5
Mazzuca et al., (1997)	Yes	Yes	Yes	Yes	No	No	No	No	Yes	Yes	Yes	6
Giraudet-Le et al., (2003)	Yes	Yes	Yes	No	No	No	Yes	Yes	Yes	Yes	Yes	7
Heuts et al., (2005)	Yes	Yes	No	Yes	No	No	Yes	No	Yes	Yes	Yes	6
Ravaud et al., (2009)	Yes	Yes	Yes	Yes	No	No	No	Yes	Yes	Yes	Yes	7
Arnold et al., (2010)	No	Yes	Yes	Yes	No	No	Yes	No	No	Yes	Yes	6
Allen et al., (2010)	Yes	Yes	No	Yes	Yes	No	Yes	Yes	Yes	Yes	Yes	8
Hansson et al., (2010)	Yes	Yes	Yes	Yes	No	No	Yes	Yes	Yes	Yes	Yes	8
Bezalel et al., (2010)	Yes	Yes	No	Yes	No	No	Yes	No	Yes	Yes	Yes	6
Arnold et al., (2011)	Yes	Yes	No	No	No	No	Yes	Yes	Yes	Yes	Yes	6
Somers et al., (2012)	Yes	Yes	No	Yes	No	No	Yes	No	Yes	Yes	Yes	6
Coleman et al., (2012)	Yes	Yes	Yes	Yes	No	No	Yes	Yes	Yes	Yes	Yes	8
Hughes et al., (2014)	Yes	Yes	No	Yes	No	No	No	No	No	Yes	Yes	4
Bennell et al., (2014)	Yes	Yes	Yes	Yes	Yes	No	Yes	Yes	Yes	Yes	Yes	9
Saraboon et al., (2015)	Yes	Yes	No	No	No	No	No	Yes	Yes	Yes	Yes	5
Bennell et al., (2018)	Yes	Yes	Yes	Yes	Yes	No	Yes	Yes	Yes	Yes	Yes	9
Lawford et al., (2018)	Yes	Yes	No	Yes	Yes	No	Yes	Yes	Yes	Yes	Yes	8
Ganji et al., (2018)	Yes	Yes	No	Yes	No	No	No	Yes	Yes	Yes	Yes	6
Lluch et al., (2018)	Yes	Yes	No	Yes	Yes	No	Yes	Yes	Yes	Yes	Yes	8
Allen et al., (2019)	Yes	Yes	Yes	Yes	No	No	No	Yes	Yes	Yes	Yes	7

**Table 2 ijerph-19-06194-t002:** Risk of bias.

Risk of Bias	Randomization Process	Deviations from Intended Interventions	Missing Outcome Data	Measurement of the Outcome	Selection of the Reported Result	Overall Bias in %	Results
Assignment to intervention (the ‘intention to-treat’ effect)							
Total number of studies = 19							
Low risk	73.7%	94.7%	89.5%	63.2%	94.7%	42.1%	8 studies
Some concerns	26.3%	5.3%	0%	21.1%	5.3%	31.6%	6 studies
High risk	0%	0%	10.5%	15.8%	0%	26.3%	5 studies
Adhering to intervention (the ‘per-protocol’ effect)							
Total number of studies = 1							
Low risk	100%	100%	100%	100%	100%	100%	1 study
Some concerns	0	0	0	0	0	0	0
High risk	0	0	0	0	0	0	0

**Table 3 ijerph-19-06194-t003:** Characteristics of included studies.

Authors	Population	Design	Assessment	Outcomes	Intervention	Results
Keefe et al., (1996)	Total: 88. Age: 62.6 (average, SD 10.1 years). Inclusion criteria: knee OA.	RCT	Pre- and post-intervention (10 weeks treatment period).	Pain: AIMS scale. Coping: Coping Strategies Questionnaire. Pain Behaviour: video.	G1: Spouse-assisted. G2: Coping skills training. CT: Arthritis education spouse-support. (2 h per week (10 weeks of treatment) to all groups)	Pain: Lower pain levels G1 vs. Control post-treatment (AIMS scale). Lower pain levels G2 vs. Control post-treatment, but not statistically significant. G1 vs. G2 did not differ significantly. Pain coping: G1 higher post-treatment scores in Coping Attempts factor than Control (*p* < 0.0005). G2 slightly higher than control (*p* < 0.43). Pain behavior: G1 had lower levels of pain behavior than control (*p* < 0.011). G2 had lower levels of pain behavior than control (*p* < 0.14), but not statistically significant.
Mazzuca et al., (1997)	Total: 211. Age: 62.8 in G1 (average, SD 12.2) and 62 in CG (average, SD 11) Inclusion criteria: OA.	CT	Pre- and post-intervention at 4 months intervals for 1 year.	Disability: HAQ. Pain: HAQ.	G1: individualized 30–60 min education. CG: attention group (20 min standardized public education presentations on arthritis).	Disability: G1 had significantly lower scores for disability than C. Pain: G1 had a significantly lower score for resting knee pain throughout the year of post-intervention follow-up.
Giraudet-Le et al., (2003)	Total: 99. Age: 62.7 (avg, SD 8.8). Inclusion criteria: hip OA.	RCT	Pre-intervention (2–6 weeks before surgery) and post-surgery (1 week after).	Anxiety: State Anxiety Inventory. Pain: VAS.	G1: Education (1 session). CT: Usual care (leaflet).	Anxiety: Better results in G1, but not statistically different. Pain: Pre-surgery and post-surgery pain were lower in G1 than CT.
Heuts et al., (2005)	Total: 273. Age: 51 (5.0) in G1 and 52.2 (5.1) in CT. Inclusion criteria: knee or hip OA.	RCT	Pre- and post-intervention (21 months).	Pain: VAS. Function: WOMAC. QoL: SF-36. Kinesiophobia: TSK.	G1: Education (6 sessions of 2 h) CT: Usual care.	G1 improved pain and WOMAC, while CT did not improve VAS and worsened WOMAC at 3 months and 21 months follow-up.
Ravaud et al., (2009)	Total: 327. Age: 64.3 (SD 8.3). Inclusion criteria: knee OA.	RCT	Pre- and post-intervention (4 and 12 months).	Pain: NRS. Function: WOMAC. Mental health: SF- 12.	G1: Education + Proposed exercise (through education). CT: Usual care.	G1 showed less pain (−1.65 NRS) than CT at 4 months. G1 showed less pain (−1.35 NRS) and function (−8.67 WOMAC) than CT at 12 months.
Allen et al., (2010)	Total: 515. Age: 60.1 (average, SD 10.4 years). Inclusion criteria: knee OA or hip OA.	RCT	Pre- and post-intervention (12 months).	Pain: AIMS scale. Physical function: AIMS2. Mood and tension: AIMS2. Self-efficacy: Arthritis Self-efficacy Scale.	G1: Self-management OA education. G2: General health education. CT: Usual care.	Pain: 0.4 lower G1 than CT, 0.6 lower G1 than G2 (AIMS). Physical function and mood/tension (AIMS2): Not statistically different. The mean AIMS2 walking and bending improved by 0.5 in G1 than G2 at 12 months. Self-efficacy: G1 was 0.4 higher than G2 or CT.
Arnold et al., (2010)	Total: 79. Age: 73.2 in G1 (average, SD 4,8), 74.4 in G2 (average, SD 7.5) and 75.8 in CG (average, SD 6.2). Inclusion criteria: hip OA, at least 1 fall risk factor and 65 years or older.	RCT	Pre- and post-intervention (after 11 weeks).	Balance: The Berg Valance Scale. Walking performance: 6-min walk. Functional perdormance: 30-s chair stand. Falls Efficacy: ABC. Dual task function: TUG. Arthritis impact: AIMS-2.	G1: acquatics exercise and education (exercise twice a week and education once a wk for 11 wks). G2: acquatics exercise (twice a week for 11 wks). CG: usual activity for 11 wks.	Arthritis impact: no significant differences between groups (*p* = 19). Falls efficacy: significant differences between groups in favour of G1. Functional performance: G1 significantly improved in number of chair stands compare with both G2 and CG. Dual task function: G1 showed more significant improvements than G2 and CG. Walking performance: G1 showed significant improvements than G2 and CG.
Hansson et al., (2010)	Total: 144. Age: 62 in G1 (average, SD 9.43) and 63 in CG (average, SD 9.51). Inclusion criteria: knee, hip, or hand OA with pain, stiffness, and limitation of ROM.	RCT	Pre- and post-intervention (at 6 months).	Self-care: EQ5D. Usual activities: EQ5D. Pain/discomfort: EQ5D. Anxiety/depression: EQ5D.	G1: Education program. Five sessions, three hours for each session, once a week for five weeks. CG: Living as usual.	G1 showed higher results after 6 months in all the parameters.
Bezalel et al., (2010)	Total: 55. Age: 74 in G1 (average, SD 5.1) and 73 in CG (average, SD 5.5). Inclusion criteria: 65 years old or older and knee OA.	RCT	Pre- and post-intervention (4 and 8 weeks).	Pain: WOMAC. Stiffness: WOMAC. Physical function: WOMAC. Physical function: Sit-to-stand test. Physical function: Get-up-and-go test.	G1: Education followed by a self-executed exercise program. Once a week for a month. Each session lasted 45 min. CG: Short wave diathermy for six 20 min sessions.	Pain: significant improvement after 4 weeks and no differences between groups. At follow-up in week 8, the study group continued to improve, while no change was reported for the CG. Physical function: significant improvement after 4 weeks and no differences between groups. At week 8, G1 continued to improve in all outcome parameters, excluding the sit-to-stand test and stiffness variables.
Arnold et al., (2011)	Total: 54. Age: >70. Inclusion criteria: hip OA.	RCT	Pre- and post-intervention (11 weeks).	Balance: Berg, Modified Test of Sensory Interaction. Function: TUG, 30-s chair stand. Walking: 6MWT. Falls: ABC scale. Activity level: PASE.	G1: Education + Exercise. (Education: 30 mins/week during 11 weeks). CT: Exercise only.	Intervention group: improvements in terms of risk of falls compared to control. Statistical changes in balance (Modified test of Sensory Interaction) and falls (ABC). Control group: No significant changes with respect to baseline.
Somers et al., (2012)	Total: 232. Age: 57.95 (avg, SD 10.41). Inclusion criteria: knee OA.	RCT	Pre- and post-intervention (6 months).	Pain: AIMS. Function: WOMAC. Catastrophizing: PCS. Self-Efficacy: Arthritis Self-Efficacy Scale.	G1: Pain coping skill training. G2: Behavioral weight management. G3: Pain coping skill + Behavioral weight management. CT: Usual care.	Pain (AIMS) and function (WOMAC): patients in G3 showed the lowest pain post-treatment, followed by G1, G2, and CT. Statistical differences between G3 and G2, but not G1. PCS: G3 showed the lowest post-treatment levels, followed by G1, CT, and G2. Statistical differences between G3 and G2, but not G1. Self-Efficacy: patients in G3 showed the lowest pain post-treatment, followed by G1, G2, and CT.
Coleman et al., (2012)	Total: 146. Age: 65 (average, SD 8). Inclusion criteria: knee OA.	RCT	Pre- and post-intervention (at 8 weeks and 6 months).	Pain: WOMAC. Physical function: WOMAC. Physical function: SF-36. Physical function: TUG. Role physical: SF-36. Body pain: SF-36. Vitality: SF-36. Social function: SF-36. Knee range of motion: goniometer. Muscle strenght: Kg.	G1: 6 weeks self-management education program on OA, six weekly sessions of 2.5 h. CG: 6 months waiting period before entering the G1 program.	Pain: significant improvement in G1 compared with C Physical function: significant improvement in G1 compared with C. Role physical: there were improvements in G1 compared with CG. Body pain: there were improvements in G1 compared with CG. Vitality: there were improvements in G1 compared with CG. Social function: there were improvements in G1 compared with CG. Knee range of motion: small increases in ROM were observed in G1 compared with C. Muscle strength: G1 showed improvements in quadriceps and hamstring strength during isometric contraction compared with C.
Hughes et al., (2014)	Total: 150. Age: 73.5 in G1 (Average, SD 6.75) and 73.7 in G2 (Average, SD 6.32). Inclusion criteria: hip or knee OA with symptoms.	RCT	Pre- and post-intervention (at 2 and 6 months).	Self-Efficacy for Arthritis Self-Management: LSES. Functional Lower Extremity Muscle Strength: timed-stand. Six-Minute Distance Walk: six-min walk test. Pain: WOMAC. Stiffness: WOMAC. Physical function: WOMAC. Exercise Adherence Self-Efficacy: McAuleye. Adherence: King classification.	G1: exercise program (range of motion, resistance, aerobic) and education problem solving regarding self-efficacy for exercise and adherence. Ninety minutes of intervention held three times per week for 8 weeks. The first 60 min of the intervention included exercise and the last 30 min included education. CG: a copy of *“the arthritis helpbook*” and a list of exercises.	Self-efficacy for Arthritis Self-management: significant difference in favor of G1. Functional lower extremity muscle strength: no significant differences. Six-minute distance walk: significant differences were seen favoring the G1. WOMAC: significant differences favoring G1 were seen in pain and stiffness. Exercise adherence self-efficacy: no significant difference. Adherence: significant differences in G1.
Bennell et al., (2014)	Total: 102. Age: 64.5 in G1 (average, SD 8.6) and 62.7 in CG (average, SD 6.4). Inclusion criteria: hip OA, Pain in groin or hip, and Age > 50.	RCT	Pre- and post-intervention (13 and 36 weeks).	Pain: VAS Function: WOMAC	G1: Manual therapy, home exercise, education and advvice. During the follow-up performation of home exercise 3 times/w. CG: Sham intervention included inactive US and inert gel on the hip region. During the follow-up, application of inert gel for 5 min for 3 times/w Physical therapy was applied over 12 weeks; twice in the first week, once weekly for 6 weeks, then approximately once every 2 weeks. The initial two sessions were 45 to 60 min in duration. The remainder was 30 min.	Pain: the between-group differences for changes in pain were not significantly. Mean difference of 6.9 mm in favour of sham therapy. Function: no between-group differences existed for physical function. Mean difference of 1.4 units in favour of sham therapy. Both groups did not show statistically significant improvements in pain and physical function.
Saraboon et al., (2015)	Total: 80. Age: 67.5 (average, SD 7.32) in G1 and 67.30 (average, SD 6.30) in CG. Inclusion criteria: knee OA with symptoms and overweight.	RCT	Pre- and post-intervention (at 8weeks).	Knee pain: NRS. Movement ability: time-up-and-go test. ROM: Goniometers measurements Weight. Perception of illness: brief illness representation. Health behavior on OA: Health behavior questionairre	G1: Health education, weight reduction program, and quads exercise training with home-based exercise program. The program was performed for 8 weeks. CG: OA knee booklet and video compact disc.	Knee pain: G1 reported less pain than C. Movement ability: G1 reported better movements than C. ROM: G1 reported better ROM than C. Weight: lower body weights after G1 than C. Perception of illness: Participants in G1 shows better result than C. Health behaviour on OA: participants in G1 showed better results than participants in CG.
Bennell et al., (2018)	Total: 144. Age: >61. Inclusion criteria: hip OA.	RCT	Pre- and post-intervention (8 weeks, 24 weeks, 52 weeks).	Pain: NRS. Function: WOMAC. QoL: AQoLv2. Catastrophizing: PCS.	G1: PCST + Education + Exercise. CT: Education + Exercise.	G1 provided no better clinical outcomes than CT in terms of pain and function at 24–52 weeks.
Lawford et al., (2018)	Total: 148. Age: >60. Inclusion criteria: knee OA.	RCT	Pre- and post-intervention (3 months, 9 months).	Pain: NRS. Function: WOMAC.	G1: Education + PCST + Internet-based exercise. CT: Education.	No differences in terms of pain except for employment patients, which patients in G1 showed less pain at 3 months follow-up. No differences between groups in terms of function at any follow-up.
Ganji et al., (2018)	Total: 82. Age: >60. Inclusion criteria: knee OA.	RCT	Pre- and post-intervention (8 weeks after).	Pain: VAS.	G1: Education. CT: Usual care. Education: 60′/2 per week, for 3 weeks.	G1 diminished pain after intervention and 8 especially 8 weeks after the intervention in comparison with the CT.
Lluch et al., (2018)	Total: 44. Inclusion criteria: Knee OA, pain of more than 3 months, and scheduled to undergo total knee replacement.	RCT	Pre-intervention, post-intervention (immediately and after 1 month), and post-surgery (3 months).	Conditioned pain modulation: Catchart protocol Pressure pain thresholds: Fisher algometers Temporal summation: Catchart protocol Symptoms of central sensitization: CSI questionnaire Knee pain: WOMAC Disability: WOMAC Physicosocial variables and pain catastrophizing: PCS Kinesiophobia: TSK-11	G1: Pain neuroscience education and knee joint mobilization. CG: Biomedical education and knee joint mobilization.	Conditioned pain modulation: only significant change was observed for the experimental treatment between baseline CMP value and the value measured at 3 months post-surgery. Temporal summation: no changes over time. Pressure pain thresholds: it did not differ between treatments but changed over time. For both treatments, there was a significant increase in PPT at all locations immediately post-treatment, at 1 month after treatment, and at 3 months after surgery. Symptoms of central sensitization: improved over time in both groups when measured at 3 months post-surgery with no difference between treatments. Pain: improvements in both groups but no differences between G1 and C. Disability: improvements in both groups but no differences between G1 and C. Physical and social variables: improvements in experimental group at 3 months post-surgery, immediately post treatment and at 1 month after treatment. Signficantly lower values of PCS were seen with the experimental compared to control treatment.
Allen et al., (2019)	Total: 248. Age: 59 (average, SD 10.3 years). Inclusion criteria: African Americans, knee or hip symptomatic OA.	RCT	Pre- and post-intervention (3 and 9 months).	Pain: WOMAC Function: WOMAC Coping strategies: CSQ Pain Catastrophizing: PCS Arthritis Self Efficacy: ASES Physical Activity Survey: YPAS Patient Global Impression of Arthritis Symptom Change	G1: Pain coping skills training. CG: usual care.	Pain: no differences between G1 and CG. Function: no differences between G1 and CG. Coping strategies: significant improvements in G1. Pain catastrophizing: significant improvements in G1. Arthritis self-efficacy: better improvements in G1. Physical activity survey: no differences between groups. Patient global impression of symptoms: better improvements in G1.

## Data Availability

The data presented in this study are available on request from the corresponding authors.

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
