# Peer review of "Effects of Patient Education on Pain and Function and Its Impact on Conservative Treatment in Elderly Patients with Pain Related to Hip and Knee Osteoarthritis: A Systematic Review"

_ijerph, 2022, doi:10.3390/ijerph19106194_

Round 1

Reviewer 1 Report

Effects of Patient Education on Pain and Function and its Impact on Conservative Treatment in Elderly Patients with Pain Related to Hip and Knee Osteoarthritis: A Systematic Review

Sinatti, P, et al.

Manuscript ID: ijerph-1698939

Summary: This manuscript reviews 1732 studies related to patient education and treatment for people with hip and knee OA. Of these studies, 20 were found to meet all of the criteria and were examined in more detail. These studies show a positive effect of education on reducing pain and improving function for patients with knee and hip pain.

Major Comments:

This manuscript is a good summary of the available literature on patient education. There are just a few major comments related to clarity of the material.

Figure 1 explaining the PRISMA flow diagram needs some better discussion in the text. The numbers don’t always correspond between the text and the figure (e.g. were 1731 or 1732 records identified). It is unclear why records are excluded in the screening or not retrieved. The number of reports excluded don’t add up to 70 so why were the 90 eligible reports reduced to 20?

Table 1 (and Table 3) are not ordered in any way that is obvious. Perhaps ordering by publication date or author would make these easier to sort through.

The entire manuscript needs to be proofread for grammar and spelling in English. There are too many mistakes currently to be acceptable in its current form. Some of these will be indicated in the minor comments, but the whole document needs to be checked thoroughly. There also seem to be extra spaces and periods in various places.

Minor Comments:

26           grammar; “… it impact on conservative …”

43           “women”

48           “patient”

64           grammar; “… but it exists no recent …”

84           grammar; “… not on others conservative …”

84-85    “… did not analyze the effects …” (no s)

86           grammar; “… positively impact in conservative … “

88           “his” ? maybe supposed to be “this”

99           grammar; “… and how it impact on conservative …”

I will stop indicating spelling and grammar errors and missing words at this point, but there are many more that must be fixed.

109         I would suggest using commas instead of hyphens.

118         specify what “elderly” is considered in the age range

158         Number doesn’t match with table (1731 vs 1732). What databases were used for the search? MEDLINE, Pubed, and Cochrane are all mentioned at different points in the manuscript.

Figure 1

                The numbers need to be checked. 1732 records screened minus 1473 records excluded (no explanation why excluded) DOES NOT leave 258 reports for retrieval. 90 reports assessed for eligibility but only 51 excluded; however, only 20 studies were included. What happened to the other studies?

Table 1  A number of the authors are not spelled correctly.

Table 2 and associated text (175 – 179). This table is confusing. I believe the numbers in the table with commas are percents (so 73.7% written as 73,7. This is confusing. Perhaps just listing the number of studies that fall into this criteria will be easier to read. The text also talks about “the remaining twelve” that are not low risk, but the numbers in the table don’t support this number. I don’t know how the statement that 6 results are “high risk” can be supported.

Table 3  The format of this table is poor and takes up too much space. Suggested reformatting to be more concise. The summaries are also confusing and somewhat inconsistent.

Reviewer 2 Report

Although the topic is interesting and clinically important, the review has been written in a chaotic way which makes it difficult to understand how it has been exactly conducted. The study aims are not clear, and the methodology is described in a superficial way. The methodology is questionable given the limited search strategy and databases used, unclear data synthesis methods, and eligibility criteria. Thus, I question the validity of the study.

ABSTRACT

     1. It is not clear what are the study aims.

  1. Please tone down your conclusions - data presented do not fully justify such conclusions.

INTRODUCTION

  1. Lines 40-43: I do not understand what is the rationale for reference 2 here?
  2. What is the need behind explaining what are the costs of OA in almost each European country?
  3. Can authors implement paragraphs in the introduction?
  4. Line 59: Complementary interventions such as knee bracing (https://pubmed.ncbi.nlm.nih.gov/30099859/ ), and physical modalities ( https://pubmed.ncbi.nlm.nih.gov/30315763/ ) should also be acknowledged.
  5. In the current form, it is quite difficult to figure out from the information flow in the introduction, why it is important to study this, who will benefit from it, and what is the added value of this paper to current knowledge. Please clarify.

METHODS

  1. Please provide PROSPERO registration number and explain and discrepancies between the protocol and this manuscript.
  2. Why did you decide to search only MEDLINE?
  3. It is not clear how you synthesized data to achieve the study objectives?
  4. Please include PRISMA checklist as an appendix.

RESULTS

  1. Figure 1: did you search articles in MEDLINE or in Pubmed/Cochrane as figure 1 suggests?
  2. Table 3 is unreadable.

DISCUSSION

  1. Please provide information on how your results will impact research and/or clinical practice.
  2. Please discuss how the generalizability of the results to the wider OA population.
  3. Limitations of the review should be discussed.

Round 2

Reviewer 2 Report

The authors fully completed my comments.